# Optimization of Process Parameters and Analysis of Microstructure and Properties of 18Ni300 by Selective Laser Melting

**DOI:** 10.3390/ma15144757

**Published:** 2022-07-07

**Authors:** Yaxin Ma, Yifei Gao, Lei Zhao, Dongling Li, Zhengxing Men

**Affiliations:** 1Central Iron & Steel Research Institute, Beijing 100081, China; 2NCS Testing Technology Co., Ltd., Beijing 100081, China; 3China Iron and Steel Research Institute Group, Beijing 100081, China; lidongling@ncschina.com; 4Chengdu Aeronautic Polytechnic, Chengdu 610100, China; amen1980@163.com

**Keywords:** selective laser melting, maraging stainless steel, microstructure, mechanical properties

## Abstract

In this research, we studied the influence of process parameters on the quality of selective laser melting of 18Ni300 maraging steel. The effects of laser power and scanning speed on the relative density and hardness of 18Ni300 were studied by single-factor experiment and the orthogonal experimental method. The relative optimal process parameters of 18Ni300 were obtained when the layer thickness was 0.03 mm, and the hatch space was 0.1 mm. The microstructures and mechanical properties of the samples formed under different process parameters were characterized. The results showed that the optimal hardness and relative density of the sample were 44.7 HRC and 99.98% when the laser power was 230 W and the scanning speed was 1100 mm/s, respectively; the microstructure of the material was uniform and dense, exhibiting few pores. Some columnar crystals appeared along the boundary of the molten pool due to vertical epitaxial growth. The orientation of fine grains at the boundary of the molten pool was random, and some coarse columnar crystals in the molten pool exhibited a certain orientational preference along the <001> orientation. In the case of optimal process parameters, the SLM-formed 18Ni300 was composed of 99.5% martensite and 0.5% retained austenite; the indentation hardness was distributed in the range of 3.2–5 GPa. The indentation modulus was between 142.8–223.4 GPa, exhibiting stronger fluctuations than the indentation hardness. The sample’s mechanical properties showed obvious anisotropy, while the tensile fracture characteristics exhibited necking. The tensile fracture morphology was ductile, and large equiaxed dimples and holes could be observed in the fiber area, accompanied by tearing characteristics.

## 1. Introduction

In contrast to traditional machining methods, additive manufacturing (AM) is based on layer-by-layer incremental manufacturing. Most of the related AM techniques use powders or wires as raw materials, which are selectively melted by a concentrated heat source and solidified in subsequent cooling to form the desired part [1,2,3]. As an advanced AM technique, rapidly developed in recent years, Selective Laser Melting (SLM) has attracted the attention of researchers. This method is suitable for producing parts with complex structures at high precision, and has the advantages of short forming cycles and high production efficiency [1,2]. The SLM technology meets the requirements for producing and applying high-performance, complex components made of contemporary and future materials, enabling efficient production, complex structures, free formation, and excellent performance.

The manufacturing of conformal waterway inserts for injection molds is currently the main field of large-scale industrial application of SLM [2,4]. For that purpose, 18Ni300 maraging steel powders are usually processed by SLM molding. Thus, the preparation of raw powder materials and the formation and heat treatment processes have been extensively investigated. Zhao et al. [5] studied the effect of void defects during the heat treatment process on the mechanical properties and corrosion behavior of 18Ni300 maraging steel manufactured by SLM. They found that the average pore size of macropores increased due to the change in stress around the pores during the phase transition occurring in the solution treatment, and the corrosion resistance of SLM 18Ni300 was negatively correlated to pore size. Ferreira et al. [6] investigated dry-sliding wear and the mechanical behavior of 18Ni300 and H13 steels manufactured by SLM. The results showed that the specific wear rate of H13 steel was two orders of magnitude lower than that of 18Ni300 steel (0.11 × 10^−7^ mm^3^/mN); moreover, the wear mechanism of 18Ni300 steel was abrasion, while the main factor in H13 steel was fatigue. Elangeswaran et al. [7] studied the fatigue behavior of 18Ni300 martensitic aging steel produced by additive manufacturing, and found that vibration finishing and sandblasting significantly improved fatigue performance, and vibration finishing was superior to sandblasting due to better surface finish. Casalino et al. [8] conducted experimental research and continuous statistical optimization of the process parameters of 18Ni300 martensitic steel formed by SLM, and found that hardness, strength, and surface roughness were positively correlated to the part’s density. Sun et al. [9] studied the tribological behavior of 18Ni300 maraging steel under high-speed dry-sliding conditions, demonstrating that the friction coefficient of the friction pair decreased with the applied load and speed. At high loads and high speeds, iron oxides on the wear surface of 18Ni300 maraging steel transformed from FeO to Fe_3_O_4_, changing the wear mechanism from adhesive wear to severe oxidation or extrusion wear. Ricardo et al. [10] investigated the low-cycle fatigue behavior of 18Ni300 maraging steel manufactured by SLM. The material exhibited slightly strain-softening behavior and nonlinear responses in elastic and plastic states. Furthermore, the transition life of this steel was very low, which could be attributed to the combination of high strength and low ductility. The total strain energy density was fairly stable over the life cycle, regardless of the strain amplitude. Rivalta et al. [11] studied the effects of scanning strategy on size, roughness, density, and hardness, indicating a significantly weaker influence of scanning strategy compared to residual stress and deformation. Unavoidable defects in the formation process, such as holes, cracks, splashes, and spheroidization, as well as the control of the forming structure, are important factors restricting the development of SLM technology [2,12].

In addition, these defects were also formed due to the differences in equipment, printing parameters, and printing environments. Therefore, the molding process parameters were optimized to reduce defect formation and to obtain a test piece with excellent performance, which is critical for an SLM forming process. To obtain high-performance 18Ni300 maraging steel and formed parts by SLM, we used a single-factor experiment and the orthogonal experimental method to study the influence of laser power and scanning speed on the density and hardness of formed parts, using a layer thickness of 0.03 mm. The optimal process parameters of 18Ni300 were obtained when the hatch space was 0.1 mm. The microstructure and mechanical properties of the samples formed by the optimal process parameters were characterized, providing a reference for the high-quality formation of 18Ni300 maraging steel.

## 2. Test Materials, Equipment, and Process Parameters

### 2.1. Test Material

The selected material was 18Ni300 maraging steel prepared by the vacuum air mist method (see Table 1). The particle size distribution of the powder was in the range of 25–53 µm, the Hall flow rate was 13.80 s/50 g, and the bulk density was 4.18 g/cm^3^. The powder morphology and particle size distribution are shown in Figure 1, illustrating mainly spherical particles with a smooth surface.

### 2.2. 18Ni300 Forming and Testing Equipment

During the test, we used a DMP Flex350GF SLM forming setup equipped with a 500 W fiber laser with a spot diameter of 65 µm. The powder was fed in two directions by a soft scraper; the powder thickness was 10–100 µm, the oxygen content was ≤25 ppm, the maximum scanning speed was 7 m/s, the maximum forming efficiency was 35 cm^3^/h, and the maximum formation size was 275 mm × 275 mm × 380 mm.

After the SLM forming of the 18Ni300 material, the sample was corroded with an aqua regia solution, and the defects were analyzed by a Bruker three-dimensional X-ray microscope SKYSCAN 2214. The microstructure in different directions was examined by a DM2700M metallographic microscope, and the density was measured by an MH-600A direct reading solid densitometer. We used an MTS E45-305 universal tensile testing machine to measure the mechanical properties and a Hitachi SU3500 scanning electron microscope for microstructural observation, fracture analysis, etc.

### 2.3. Experimental Scheme

The experimental plan was completed in three stages to obtain the best formation process for 18Ni300. To determine the scope of the best process plan, according to previous experience with the equipment we selected a layer thickness of 0.03 mm and a hatch space of 0.1 mm. The laser power in the orthogonal process parameter experiments was adjusted to 210, 220, 230, 240, and 250 W, and the scanning speed was 900, 1000, 1100, 1200, and 1300 mm/s for 25 groups of 10 mm × 10 mm rectangular specimens. After forming, the density, hardness, and other parameters of the 18Ni300 specimens were analyzed to determine the basic parameter range. The best process parameters were determined using orthogonal test range analysis. The sample was printed under the best process parameters and its microstructural and mechanical properties were characterized.

## 3. Experimental Results and Analysis

### 3.1. Density Analysis

Figure 2 shows the relative density of 18Ni300 obtained by SLM forming as a function of the scanning speed and laser power. At different laser powers and scanning speeds, the maximum relative density of the material reached 99.98%, while the minimum relative density was 95.85%. In particular, the density of the 18Ni300 material first increased and then decreased with the scanning speed, and it initially rose and then dropped with the laser power. When the laser power was 230 W and the scanning speed was 1000 and 1100 mm/s, the obtained relative density of the samples was highest.

To study the reasons for the relative density difference, three samples with different printing parameters were selected for partial micro-CT analysis to determine the porosity distribution in the samples. A Bruker three-dimensional X-ray microscope SKYSCAN 2214 was used; the source voltage was 130 kV, and the source current was 35 µA. The image pixel size was 3 µm, and the test results are shown in Figure 3, where the a-1, b-1, and c-1 panels of Figure 3 represent the 2D distribution of pores, and the a-2, b-2, and c-2 panels of Figure 3 show the 3D distribution of pores. When the laser power was 210 W, and the scanning speed was 1300 mm/s, shown in Figure 3a, there were many pores of different shapes and sizes; when the laser power was 210 W, and the scanning speed 1100 mm/s, shown in Figure 3b, the number of large pores decreased, and the distribution of small pores was uniform; when the laser power was 230 W, and the scanning speed was 1100 mm/s, shown in Figure 3c, there were only a few small pores. Therefore, when the hatch space was 0.1 mm, and the layer thickness was 0.03 mm, the laser power and scanning speed were closely related to pore size, distribution, shape, and position. In addition, because the resolution was set to 3 µm, all pores analyzed in this experiment were larger than 3 µm, and those smaller than or close to 3 µm were undetectable. Thus, the experimentally determined porosity may be lower than the actual if the pores were ≤3 µm.

### 3.2. Hardness Analysis

The hardness of the 18Ni300 sample manufactured by SLM under different laser powers and scanning speeds is shown in Figure 4. The hardness of the sample manufactured by SLM increased first and then decreased with the scanning speed. As the scanning speed increased, the dispersion of hardness became significant; when the scanning speed was in the low and medium speed range (900–1100 mm/s), the hardness of samples increased with the laser power. Globally, the hardness first increased and then decreased. In high-speed scanning (1200–1300 mm/s), the hardness did not show characteristic behavior with the laser power. In general, the hardness was higher for a laser power of 240 W and a scanning speed of 1000–1100 mm/s.

### 3.3. Orthogonal Test and Range Analysis

The orthogonal test and range analysis were carried out for the above tests and results. The orthogonal experimental design and the results of the relative density and hardness are shown in Table 2. The range analysis of the orthogonal test is shown in Table 3. The results indicate that the relative density exhibited different sensitivities to the investigated experimental parameters, arranged in the order of laser power > scanning speed, while the hardness sensitivity is arranged in the order of scanning speed > laser power. These results corroborate those obtained in the single factor experiment. According to the orthogonal test and range analysis results, the optimal formation conditions were achieved when the layer thickness was 0.03 mm and the hatch space was 0.1 mm at a laser power of 230 W and a scanning speed of 1100 mm/s. Using these parameters to print the sample, the relative density was 99.98%, which was better than that for other samples; the hardness was 44.7 HRC, which is relatively high and better than values reported in the literature [8,13].

### 3.4. Microstructure and Micromechanical Properties

Metallographic polishing (perpendicular to the laser beam direction) was performed on the SLM samples obtained under the optimal molding conditions. The deposition morphology of the molten pool after corrosion is shown in Figure 5a. The depth of the molten pool was about 30 µm, which is consistent with the layer thickness of the powder. The width of the pool was about 100 µm, consistent with the hatch space, and only a few holes appeared, which is related to the printing parameters, environment, and powder quality during laser forming [3,7,13]. From the high-magnification SEM image in Figure 5b, it can be seen that the rapid cooling of powder after laser melting additionally densified the sample microstructure, yielding fine columnar, dendritic, and cellular structures and replacing the traditional lath and massive martensitic structures, mainly due to the segregation and aggregation of some solute elements (Ni, Mo, Ti) in the cellular structure during the rapid solidification of powder after laser melting. The microcellular structure was uniform and dense and grew along the direction of thermal diffusion. At the same time, some columnar crystals grew vertically and epitaxially along the boundary of the molten pool, which was mainly affected by the solidification rate and temperature gradient. In the laser cladding process, Wang et al. [14] found that a dendritic structure was formed when the solidification rate gradually increased, while a cellular structure appeared when the value of the temperature gradient solidification rate decreased. In addition, Figure 6a shows the grain orientation diagram of the printed state. The orientation of the fine grains at the boundary of the molten pool was random, and the orientation of the larger columnar grains in the molten pool showed a certain preference along <001>. Figure 6b shows that the sample consisted of 99.5% martensite body-centered cubic phase, in agreement with the literature analysis [3,15], while 0.5% was a face-centered cubic phase, i.e., residual austenite. For the samples as printed, solute elements may have segregated at the molten pool boundary and grain boundaries when the molten metal pool solidified. As a result, the martensitic transformation was hindered, part of the retained austenite segregated at the grain boundaries and martensite lath boundaries, and the residual austenite content is relatively high.

To further study the micromechanical properties, a G200X nanoindenter was used to perform a mapping (rapid indentation) test on the samples obtained under optimal molding conditions. The test area was 200 µm × 200 µm, the indentation depth was 300 nm, and the indentation spacing was 10 µm. Figure 7 shows the surface distribution diagram of the indentation hardness and modulus of the sample, in which the indentation hardness was uniformly distributed between 3.2–5 GPa, with an average value of about 4.2 GPa. The elastic modulus was distributed between 142.8–223.4 GPa, with an average value of about 195 GPa, and the elastic modulus fluctuated more than the indentation hardness, which may be related to the grain orientation and grain boundary distribution at the indentation position.

### 3.5. Tensile Test and Fracture Analysis

For the two types of samples obtained under optimal molding conditions (the tensile direction of the XY-sample was perpendicular to the printing direction, and the tensile direction of the Z-sample was parallel to the printing direction), the tensile test was carried out according to GB/228.1. The obtained engineering stress–strain curve is shown in Figure 8. The strength of the XY-sample was significantly higher than that of the Z-sample, but the elongation of the XY-sample was significantly lower than that of the Z-sample; that is, in different printing directions, the mechanical properties of the sample exhibited obvious anisotropy, which is attributed to the strong crystallographic texture or inhomogeneous structure of metallic parts during AM [16]. In addition, no obvious work hardening process occurred from the elastic stage to the yield point, and the strength did not increase significantly with subsequent stretching, i.e., the yield strength and tensile strength of specimens in the same direction were similar.

Figure 9 shows the tensile fracture morphology of the XY sample. In Figure 9a, necking can be observed. The fracture shows typical fiber areas, radial areas, and shear lips. There were many large dimples and certain dimples in the fiber area. The appearance of pores indicates that the sample underwent obvious plastic deformation during the tensile process, and the fracture occurred as a ductile fracture. In the fiber area, shown in Figure 9b, obvious equiaxed large dimples and holes appeared, accompanied by tearing characteristics, in which small dimples were distributed on large dimples. Figure 9c shows a high magnification of the lip shearing with small equiaxed dimples. In the 3D printing process, due to the influence of purity, uniformity, forming environment, and parameters of the powder, defects such as incomplete melting, spheroidization, and pore formation of the powder may be caused [17,18,19]. These defects can easily damage micropores under the action of tensile stress, causing their nucleation, aggregation, and growth, finally causing fracture of the sample.

## 4. Conclusions

In this paper, the influence of laser power and scanning speed on the SLM forming quality of 18Ni300 maraging steel was studied to obtain formed parts with high formation quality. The effects of laser power and scanning speed on the density and hardness of formed parts were studied using a single-factor experiment and the orthogonal experimental method. The relative optimal process parameters of 18Ni300 were obtained, and the microstructure and mechanical properties of the samples formed by the optimal process parameters were characterized. The results are summarized as follows:According to the single-factor experiment, the orthogonal experimental method, and the range analysis results, a laser power of 230 W and a scanning speed of 1100 mm/s were the optimal forming conditions when the layer thickness was 0.03 mm, and the hatch space was 0.1 m. Using these parameters to print the sample, a relative density of 99.98%, which was better than other samples, and a relatively high hardness of 44.7 HRC were obtained.After the sample was polished and corroded, the deposition morphology of the molten pool was assessed. The depth of the molten pool was about 30 µm, and the width of the molten pool was about 100 µm, which was consistent with the layer thickness of the powder and the hatch space. The microcellular structure was uniform and dense. The orientation of the fine grains at the edge of the molten pool was random, and the orientation of the larger columnar grains in the molten pool exhibited a certain preference along the <001> plane. The sample consisted of 99.5% martensite and 0.5% austenite. The indentation hardness was between 3.2–5 GPa, with an average value of about 4.2 GPa. The indentation elastic modulus was between 142.8–223.4 GPa, with an average value of about 195 GPa, and the elastic modulus fluctuated more than the indentation hardness.The strength of the XY-specimen was significantly higher than that of the Z-specimen, but the elongation of the Z-specimen was higher than that of the XY-specimen, i.e., in different printing directions, the mechanical properties of the sample exhibited obvious anisotropy. The tensile fracture morphology showed necking, and ductile fractures appeared. Lare equiaxed dimples and holes were observed in the fibrous area, accompanied by tear characteristics in which small dimples were distributed on the large dimples.

## Figures and Tables

**Figure 1 materials-15-04757-f001:**
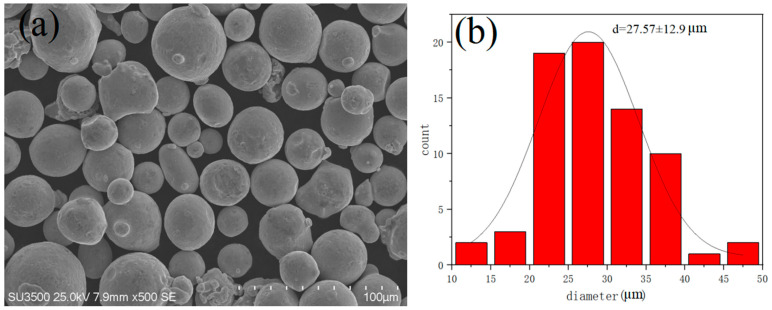
(**a**) Morphology and (**b**) particle diameter distribution of the 18Ni300 powder.

**Figure 2 materials-15-04757-f002:**
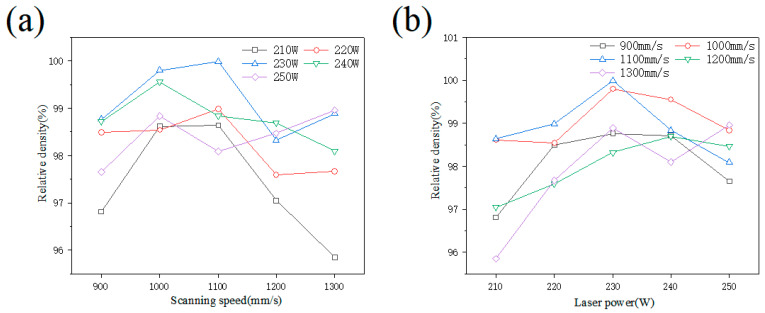
Variation in the sample’s relative density with the (**a**) scanning speed and (**b**) laser power.

**Figure 3 materials-15-04757-f003:**
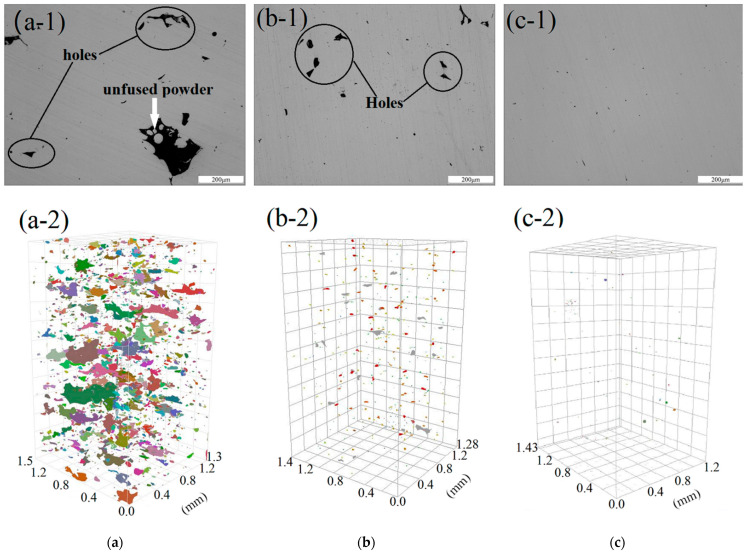
Relative density obtained for different printing parameters ((**a**) 210 W, 1300 mm/s, (a-1) 2D, (a-2) 3D; (**b**) 210 W, 900 mm/s, (b-1) 2D, (b-2) 3D; (**c**) 230 W, 1100 mm/s, (c-1) 2D, (c-2) 3D).

**Figure 4 materials-15-04757-f004:**
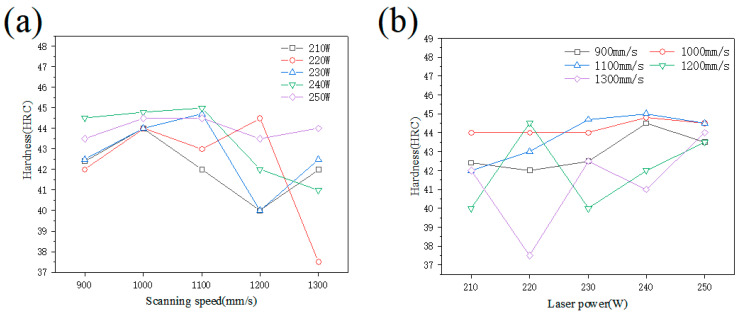
Variation of sample hardness with (**a**) scanning speed and (**b**) laser power.

**Figure 5 materials-15-04757-f005:**
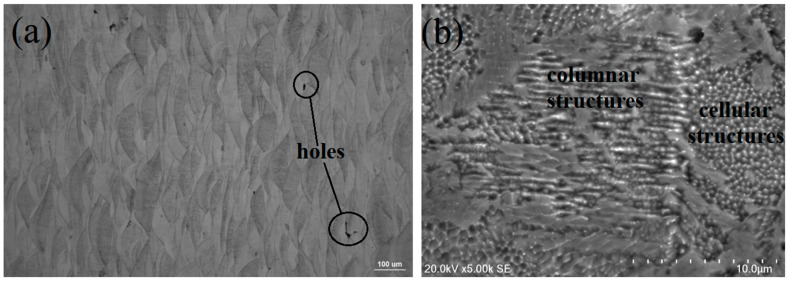
Microstructure of the SLM sample. ((**a**) OM, (**b**) SEM).

**Figure 6 materials-15-04757-f006:**
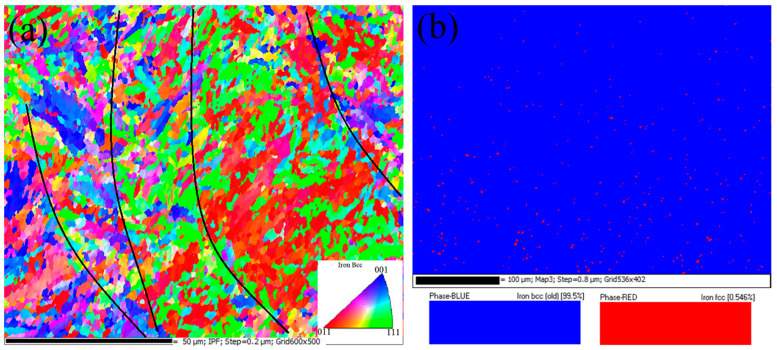
The EBSD (**a**) orientation and (**b**) phase distribution of the SLM sample.

**Figure 7 materials-15-04757-f007:**
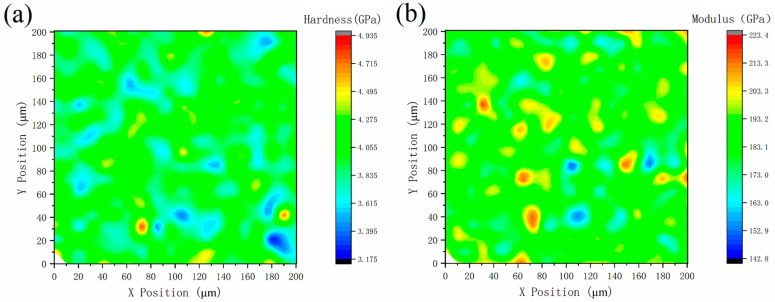
Surface distribution diagram of the indentation (**a**) hardness and (**b**) modulus of the SLM sample.

**Figure 8 materials-15-04757-f008:**
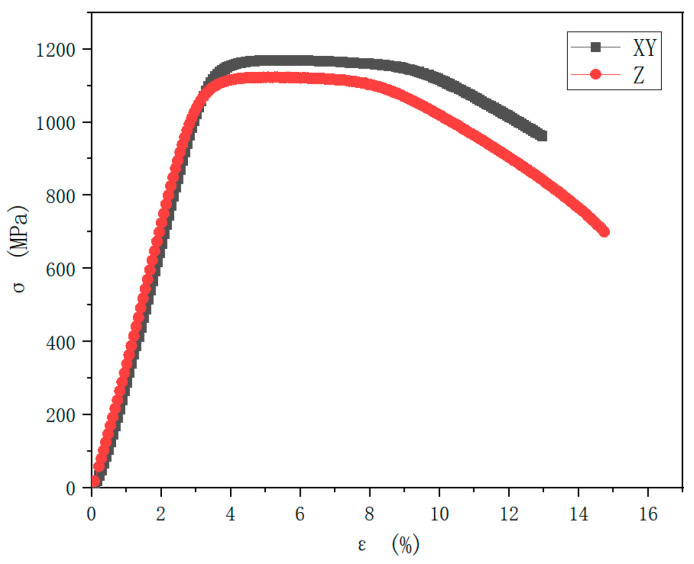
The engineering stress–strain curve of the SLM specimen.

**Figure 9 materials-15-04757-f009:**
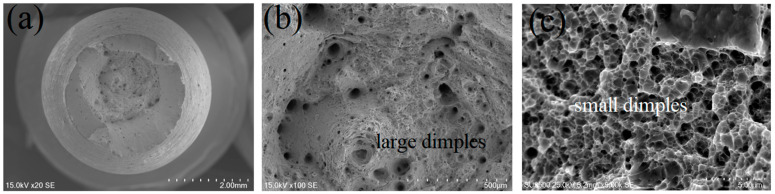
Fracture morphology of the SLM sample. ((**a**) fracture morphology, (**b**) fiber area, (**c**) a high magnification of the lip shearing).

**Table 1 materials-15-04757-t001:** Chemical composition of the 18Ni300 powder (mass%).

Ni	Ti	Co	Al	Mo	Si	Cr	Mn	C	Fe
17.70	0.72	9.05	0.077	4.79	0.025	0.031	0.022	0.007	Balance

**Table 2 materials-15-04757-t002:** Design and results of the orthogonal experiments.

Number	Laser Power(W)	Scan Speed (mm/s)	Relative Density (%)	Hardness (HRC)
1	1 (210)	1 (900)	96.81481	42.4
2	1	2 (1000)	98.61728	44
3	1	3 (1100)	98.64198	42
4	1	4 (1200)	97.04938	40
5	1	5 (1300)	95.85185	42
6	2 (220)	1	99.49383	42
7	2	2	98.54321	44
8	2	3	98.98765	43
9	2	4	97.59259	44.5
10	2	5	97.66667	37
11	3 (230)	1	98.76543	42.5
12	3	2	99.80247	44
13	3	3	99.98765	44.7
14	3	4	98.32099	40
15	3	5	98.88889	42.5
16	4 (240)	1	98.71605	44.5
17	4	2	99.55556	44.8
18	4	3	98.83951	45
19	4	4	98.69136	42
20	4	5	98.09877	41
21	5 (250)	1	99.65432	43.5
22	5	2	98.83951	44.5
23	5	3	98.08642	44.5
24	5	4	98.46914	43.5
25	5	5	98.96296	44

**Table 3 materials-15-04757-t003:** Range analysis of the orthogonal experiments.

	Factor	Relative Density (%)		Hardness (HRC)	
Project		Laser Power	Scan Spee	Laser Power	Scan Spee
Mean 1		97.395	98.689	42.08	42.98
Mean 2		98.457	99.072	42.1	44.26
Mean 3		99.153	98.909	42.74	43.84
Mean 4		98.78	98.025	43.46	42
Mean 5		98.802	97.894	44	41.3
Range		1.758	1.178	1.92	2.96

## Data Availability

All data included in this study are available upon request by contact with the corresponding author.

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
