# Peer review of "Optimization of Process Parameters and Analysis of Microstructure and Properties of 18Ni300 by Selective Laser Melting"

_materials, 2022, doi:10.3390/ma15144757_

Round 1
Reviewer 1 Report
1- Language of the paper should be checked and improved
Please review your paper to improve the English. Like the below heading should be:
Table 2. Design and results of the orthogonal experiments
2- Effect of laser power and scan speed on density and hardness seem to be minimal and within experimental accuracy. Please explain your tolerance on experimental accuracy and repeatability first, then evaluate your results accordingly in your conclusion.
Reviewer 2 Report
In general, the quality of graphs on figures must be improved.
I found that the phrase on the abstract:
"The effects of laser power and scanning speed on the relative density and hardness of 18Ni300 were studied by the orthogonal experiment method, and it was found that the optimal process parameters of 18Ni300 were obtained when the layer thickness was 0.03 mm and the hatch space was 0.1 mm"
It isn't apparent. It must be improved
Maybe the experiment design could be put before than density and hardness analysis.
Comments of results on page 7 must be related better to Figures 5, 6, and 7. For example, where are the columnar grains in Figure 5?. The question arises if they are primary dendrite arms instead of columnar grains, which must be clarified.
Scales of pictures in Figures 6 and 7 must be put in the captions.
Reviewer 3 Report
The paper presents images of very poor quality. Better quality images need to be inserted to make them more understandable.
In Chapter 3.5. state that the mechanical properties of the material show obvious anisotropy. Can you explain why anisotropy occurs and what is its impact on the analyzed material. What this means for the application of the material itself in an engineering sense.
The conclusions are vague. Conclusions need to be defined in more detail and precision.
Reviewer 4 Report
see the attached file

Round 2
Reviewer 3 Report
The authors have significantly improved the paper. There are still small flaws that could be fixed. The paper can be published in a journal.
Reviewer 4 Report
After the corrections made I propose to accept the paper.